# Rapid response to hemorrhagic fever emergence in Guinea: community-based systems can enhance engagement and sustainability

Saa André Tolno[1,2,3], Séverine Thys[3,4], Alpha Kabinet Keita[5,6], Maxime Tesch[1,3], Chloé Bâtie[1,3,7¤], Véronique Chevalier[3,8☯], Marie-Marie Olive[1,3☯]*

1 CIRAD, UMR ASTRE, Montpellier, France, 2 Higher Institute of Science and Veterinary Medicine (ISSMV), Dalaba, Republic of Guinea, 3 ASTRE, Univ Montpellier, CIRAD, INRAE, Montpellier, France, 4 CIRAD, UMR ASTRE, Vientiane, Lao PDR, 5 Guinean Centre for Training and Research in Infectious Diseases (CERFIG), Campus Universitaire Hadja Mafory Bangoura, Donka, Conakry, Republic of Guinea, 6 Translational Research on HIV and Endemic and Emerging Infectious Diseases, French Research Institute for Sustainable Development (IRD), University of Montpellier, Montpellier, France, 7 Johns Hopkins Center for Health Security, Johns Hopkins Bloomberg School of Public Health, Baltimore, Maryland, United States of America, 8 CIRAD, UMR ASTRE, Antananarivo, Madagascar

☯ Equal contributions
¤ Current address: Johns Hopkins Center for Health Security, Johns Hopkins Bloomberg School of Public Health, Baltimore, Maryland, United States of America
* marie-marie.olive@cirad.fr

## Abstract

Since the 2013–2014 Ebola virus disease outbreak, Guinea has faced recurrent epidemics of viral hemorrhagic fevers. Although the country has learned from these epidemics by improving its disease surveillance and investigation capacities, local authorities and stakeholders, including community actors, are not sufficiently involved in the disease-emergence response. As a result, measures are not fully understood and have failed to engage local stakeholders. However, recent research has shown community-based response measures to be effective. For this study, we used a qualitative participatory research approach to (i) describe and analyze the health signals that alert local stakeholders to a problem, (ii) describe the outbreak response measures implemented in Guinée Forestière from local to national levels, and (iii) identify obstacles and levers for implementing responses adapted to the local sociocultural context. Local stakeholders receive a variety of health, environmental, and sociopolitical signals. When dealing with health signals, their next step should be to follow a flowchart developed using a top-down approach and disseminated by national stakeholders. However, our interviews revealed that local stakeholders found this official flowchart difficult to understand. To address this issue, we used a bottom-up approach to co-construct with local stakeholders a response flowchart based on their perceptions and experiences. The resulting diagram opens the door to the development of a community-based response. We then identified six main obstacle categories from the interviews, including insufficient logistical and financial resources, lack

**Data availability statement:** Data cannot be shared publicly. Despite the efforts to anonymize the data, the nature of the information provided—combined with sectorial and institutionnal level details —could potentially lead to the identification of individual participants. Data would be available from the Guinean NATIONAL ETHICS COMMITTEE FOR HEALTH RESEARCH (CNERS, https://cners-guinee.org/, contact via dalphahm@yahoo.fr and oumou45@yahoo.fr) for researchers who meet the criteria for access to confidential data.

**Funding:** This work was supported by the European Union under the Agreement FOOD/2016/379-660, for the implementation of the Action EBO-SURSY "Capacity building and surveillance for Ebola Virus Disease" (https://rr-africa.oie.int/en/projects/ebo-sursy-en/). Saa André Tolno received a PhD fellowship from the Service de Coopération et d'Action culturelle (SCAC) of the French Embassy in Guinea. He was also supported by AfriCam project, funded by the French Development Agency (AFD) as part of PREACTS (PREZODE in action in the global South) program. The funders had no role in study design, data collection and analysis, decision to publish, or preparation of the manuscript.

**Competing interests:** The authors have declared that no competing interests exist.

of legitimacy of community workers, and inadequate coordination. Based on these obstacles, we suggest ways to develop a response to emerging zoonotic diseases that would enable local stakeholders to better understand their roles and responsibilities and improve their commitment to the outbreak response. Ultimately, this study should help to build an integrated, community-based early warning and response system in Guinée Forestière.

## Introduction

For over 10 years, Guinea has faced recurrent epidemics of viral hemorrhagic fevers (VHFs), and more specifically Ebola virus disease (EVD), Marburg virus disease, and Lassa fever. The emergence of VHFs, in particular the 2013–2016 Ebola virus disease (EVD) epidemic, was a health crisis of unprecedented proportions [1,2]. In 2021, two new epidemics were declared in the Guinée Forestière region (all administrative place names are not translated in English for clarity's sake): one involved the re-emergence of EVD, which started in Gouécké (N'Zérékoré prefecture) and had 16 reported confirmed cases (12 of whom died), while the second involved Marburg virus disease in Temessadou M'Boké (Guéckédou prefecture), with one reported death [3,4]. Cases of Lassa fever have also been reported in recent years in Guinée Forestière [5,6].

Guinea has drawn lessons from these epidemics by improving its disease surveillance, investigation, and response capacities. For example, the Guinean National Health Safety Agency (ANSS) developed its "Response Plan to the Ebola Virus Disease Epidemic" in 2021, which has improved the response to epidemics in the country by (i) strengthening infection prevention and control measures, (ii) providing health facilities and epidemiological treatment centers (CTEpi) with supplies (e.g., medicines, disinfectants, personal protective equipment), (iii) organizing systematic follow-up of case contacts and their care, (iv) vaccinating case contacts and at-risk healthcare workers, and (v) strengthening cross-border collaboration and surveillance at entry points [7]. Investigations of health signals have been sped up thanks to the construction of several specialized laboratories in Guinea's capital city, Conakry, and local laboratories dedicated to the early detection and confirmation of cases located as close as possible to communities in high-risk areas [7,8]. However, despite this new response plan, inadequate coordination with international and national institutions involved in health-crisis management leads to the duplication of both resources and activities during epidemics [9].

VHFs are considered zoonotic diseases, and wildlife is thought to be a reservoir for certain viruses such as the Lassa and Ebola viruses. Transmissions of pathogens of concern at the human–animal interface primarily affect rural communities, which are frequently in contact with domestic animals and wildlife and thus at the frontline of VHF emergence. VHF emergence in forested areas is also shaped by the interaction of climate, socioeconomic, and ecological dynamics. Because these dynamics are nonlinear, they make the emergence of VHFs such as EVD complex

and difficult to predict [10]. While local people are generally the most familiar with and knowledgeable about changes in the forest and fauna, emergence studies have unfortunately not always capitalized on this local expertise [11]. Local authorities and stakeholders are also not sufficiently involved in surveillance and response, which leads to response activities that are not well adapted to the local context [12]. Poor adaptation combined with lack of trust between local communities, officials, and medical professionals can give rise to protests that may occasionally turn violent as well as failure among community members to comply with response measures. For example, during the 2013–2016 Ebola epidemic, affected communities often did not abide by measures such as safe burials, or the declaration of community cases [13,14]. Community and local stakeholder engagement is now commonly regarded as a crucial entry point for gaining access and securing trust during humanitarian emergencies [15–18]. However, Le Marcis et al. [19] emphasized the importance of recognizing that within communities, power and legitimacy are always contested resources, which means that when it comes to community engagement tactics employed during emergencies, one-size-fits-all, inflexible, and top-down responses are unsuitable [19]. Community-based response measures have been shown to be effective, such as during the 2018–2020 Ebola epidemic in the Democratic Republic of the Congo, where a community-based contact isolation strategy was implemented [16].

In 2019, a study conducted in Guinée Forestière by Guenin et al. [20] exploring the community's capacity to detect emerging zoonoses and surveillance network opportunities showed that the response to disease emergence first relies on the surveillance system in place. The system's ability to detect abnormal animal or human health events at the community level depends on the capacity for early detection and rapid response to emergencies, which is based on the diversity of local knowledge of existing diseases and on recognition of clinical signs. The same survey also showed that local authorities, local staff, and communities were not sufficiently involved in drawing up intervention, surveillance, and response plans [20].

Given this information, the bottom-up approach of co-constructing alert responses to zoonotic emergence events involving different surveillance stakeholders could help them better engage with and take ownership of the system [21]. Once a health signal is observed, the success of the response depends on the type of alert and the degree to which stakeholders adhere to the response plan [22,23]. As such, a better understanding and consideration of the priorities, constraints, and levers of local and national stakeholders is needed to adapt the response system, improve its acceptability by stakeholders, and ultimately improve the system's ability to rapidly detect and control any emergence event.

Qualitative and participatory approaches are effective tools to address complex health issues by considering individual characteristics and societal influence on health determinants. In particular, these methods increase researchers' and decision-makers' abilities to consider and understand the complexity of stakeholders' behavior [24]. They are increasingly used by interdisciplinary teams to enhance stakeholders' involvement from various sectors, all embedded within a particular sociocultural context [24]. Of the various participatory approaches, participatory epidemiology is widely used to improve human and animal disease surveillance [25,26]. It is based on the collection of qualitative and semi-quantitative epidemiological data in communities through interviews and visual tools, among other methods [27,28]. Knowledge and experiences of relevant stakeholders are shared with the research team, leading to stakeholder involvement. The flexible and stimulating corpus of methods available in participatory epidemiology enable researchers to develop intervention and monitoring strategies tailored to the communities involved, considering their socioeconomic and cultural constraints [26]. Qualitative research has also been successfully used to provide baseline information and identify strategies to develop community-based responses to Ebola in Liberia [15].

By using a qualitative participatory research approach, this study aimed to (i) describe and analyze the health signals that alert local stakeholders to a problem, (ii) describe the outbreak response measures implemented at local and national level in Guinée Forestière, and (iii) identify the obstacles and levers for implementing plans adapted to the local sociocultural context and the needs of the stakeholders involved in the response. These specific objectives should ultimately help in building an integrated, community-based early warning and response system in Guinée Forestière.

## Materials and methods

### Study sites

The study areas included sites at local level in the Forest Region in southeastern Guinea (prefectures of Guéckédou and N'Zérékoré) and at national level in the capital city of Conakry (Fig 1). We selected a total of six sites within Guinée Forestière. Four sites were selected in the Guéckédou prefecture: Guéckédou town, Koundou subprefecture, Temessadou Djigbo subprefecture, and Temessadou M'Boké village (located in Temessadou Djigbo subprefecture). Two sites were selected in the N'Zérékoré prefecture: N'Zérékoré town and Gouécké village (located in the Gouécké subprefecture). These sites were chosen because they had been affected by the Ebola epidemic in 2014 and 2021, the 2021 Marburg epidemic, or sporadic Lassa fever outbreaks [1–6].

### Study design

First, we referred to bibliographic references and official documents to produce a list of stakeholders and administrative structures involved in the surveillance and response to zoonotic disease outbreaks in Guinée Forestière [9,20,29]. Additional stakeholders were then selected and introduced into the study using the snowball sampling method [30]. We used the convenient sampling method based on the different categories (but not numbers) of actors we aimed to represent at each site and level. The sample size was obtained when data saturation was reached among our studied population.

Based on this theoretical data saturation and depending on the availability of targeted stakeholders [31], 13 focus group discussions (FGDs), 13 individual in-depth interviews (IDIs), and informal discussions were carried out, involving a total of 158 participants, including 129 men and 34 women (Table 1).

In our study framework, we defined (i) an **event** as something that may occur in the community and that may have a negative impact on the community [32], (ii) a **signal** as an immediate alert at the early stages of the event (disease outbreak or disaster) that requires an immediate notification and investigation for verification (https://www.emro.who.int/health-topics/ewarn/index.html), and (iii) a **response** as actions triggered to stop or limit the consequences of the event.

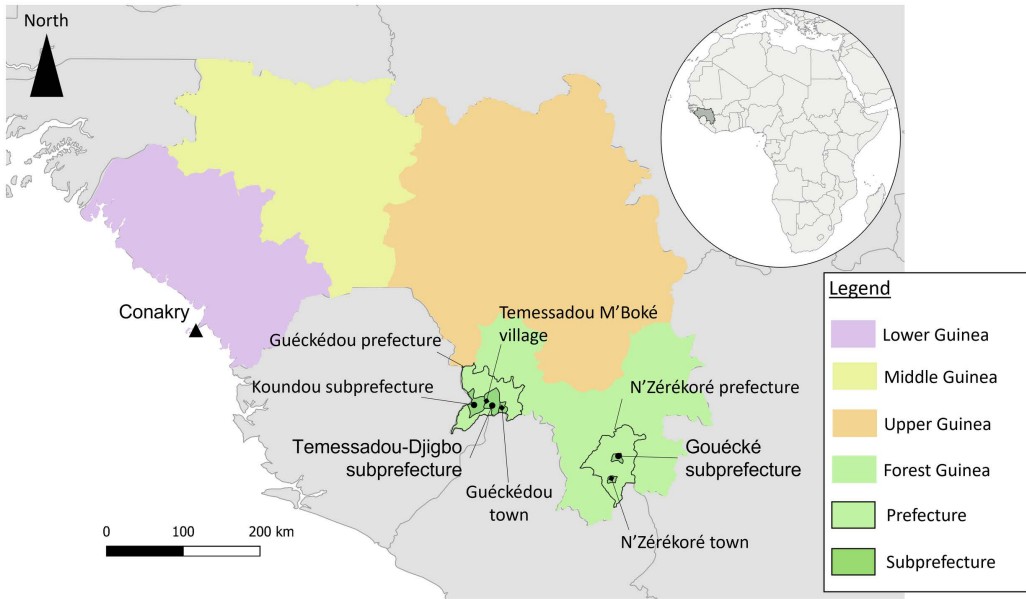

**Fig 1. Map of the study sites in Guinea.** (Map created with QGIS software: source of administrative boundaries map layer: https://www.gadm.org/; Link to the GADM license: https://www.gadm.org/license.html).

**Table 1. Characteristics of the groups involved in focus group discussions (FGDs) or individual in-depth interviews (IDIs).**

| Type | Category of actors | Level of decision | Group size M | F | Locations | Execution phase |
|---|---|---|---|---|---|---|
| FGD | Community members | Local or subprefectural | 6 | 4 | Temessadou Djigbo | Test |
| IDI | Subprefectural technical staff – Livestock | Local or subprefectural | 1 | – | | |
| IDI | Subprefectural technical staff –Human health | Local or subprefectural | 1 | – | | |
| IDI | Prefectural technical service staff – Livestock | Prefectural | 1 | – | N'Zérékoré town | Phase I |
| IDI | Prefectural technical service staff – Livestock | Prefectural | | 1 | | |
| IDI | Prefectural technical service staff – Environment | Prefectural | 1 | – | | |
| IDI | Prefectural technical service staff – Human Health | Prefectural | 1 | – | | |
| IDI | Subprefectural technical staff – Livestock | Local or subprefectural | 1 | – | Gouécké | |
| IDI | Subprefectural technical staff –Environment | Local or subprefectural | 1 | – | | |
| IDI | Subprefectural technical staff – Livestock | Local or subprefectural | – | 1 | | |
| FGD | CWs – Livestock | Local or subprefectural | 6 | – | | |
| FGD | CWs – Environment | Local or subprefectural | 7 | – | | |
| IDI | Subprefectural technical staff –Human Health | Local or subprefectural | 1 | – | | |
| FGD | CWs – Human Health | Local or subprefectural | 6 | 2 | | |
| IDI | Prefectural technical service staff – Livestock | Prefectural | 1 | | Guéckédou town | |
| IDI | Prefectural technical service staff – Environment | Prefectural | 1 | – | | |
| IDI | Prefectural technical service staff – Human Health | Prefectural | 1 | – | | |
| IDI | Prefectural technical service staff – Human Health | Prefectural | 1 | – | | |
| IDI | Subprefectural technical staff – Livestock | Local or subprefectural | 1 | – | Temessadou Djigbo | |
| IDI | Subprefectural technical staff – Environment | Local or subprefectural | 1 | – | | |
| IDI | Subprefectural technical staff – Human Health | Local or subprefectural | – | 1 | | |
| FGD | CWs – Livestock | Local or subprefectural | 5 | 1 | | |
| FGD | CWs – Human Health | Local or subprefectural | 7 | 1 | | |
| FGD | Community members | Local or subprefectural | 9 | 1 | Temessadou M'Boké | |
| FGD | CWs – Livestock and Environment | Local or subprefectural | 9 | 1 | Koundou | |
| FGD | CWs – Human Health | Local or subprefectural | 8 | 3 | | |
| IDI | Subprefectural technical staff – Human Health | Local or subprefectural | 1 | – | | |
| IDI | Subprefectural technical staff – Livestock | Local or subprefectural | 1 | – | | |
| IDI | Sub-prefectural technical staff – Environment | Local or subprefectural | 1 | – | | |
| IDI | National service staff – Livestock | National | 1 | – | Conakry | Phase II |
| IDI | National service staff – Human Health | National | 1 | – | | |
| IDI | National service staff – One Health Committee | National | 1 | – | | |
| FGD | Community members (health matrons) | Local or subprefectural | – | 9 | Temessadou Djigbo | Phase III |
| FGD | Community members | Local or subprefectural | 12 | – | | |
| FGD | CWs – Human Health | Local or subprefectural | 5 | 2 | | |
| FGD | CWs – Livestock and CWs – Environment | Local or subprefectural | 6 | 1 | | |
| FGD | Technical staff (Human Health, Livestock and Environment) | Local or subprefectural | 5 | 1 | | |
| FGD | Community members | Local or subprefectural | 7 | 4 | Koundou | |
| FGD | CWs – Human Health | Local or subprefectural | 5 | 1 | | |
| FGD | CWs – Livestock and CWs – Environment | Local or subprefectural | 6 | – | | |
| **TOTAL** | | | **129** | **34** | | |

**Legend:** Phase I = First phase of data collection in Guinée Forestière in the Temessadou-Djigbo, Koundou and Gouécké subprefectures; Phase II = Second phase of data collection in Conakry; Phase III = Third phase of data collection, feedback, and validation of preliminary results in the Temessadou-Djigbo and Koundou subprefectures; IDI = individual in-depth interview; FGD = focus discussion group; CWs = community workers; - = None.

In the context of infectious disease, a response is a set of actions "triggered to stop the spread of an infectious disease swiftly, keeping as few people as possible from being infected" (https://www.taskforce.org/outbreak-response/). In line with these definitions, we defined the process leading to a response as follows: it begins with the detection of a *signal* that alerts local stakeholders that an *event* may occur in this population, requiring actions in *response*.

## Data collection

Field work was carried out between April 2022 and April 2023 at the abovementioned sites during three separate phases. The first two phases were focused on data collection while the last phase was a validation step. The first phase was conducted from April 22, 2022, to June 25, 2022, and included a total of 91 local participants in the prefectures of Guéckédou (prefectural services in Guéckédou town and subprefecture, community workers, and community members in Temessadou Djigbo, Temessadou M'Boké, and Koundou) and N'Zérékoré (prefectural services in N'Zérékoré town and subprefecture, community workers and community members in Gouécké; Phase I, Table 1). The second phase was carried out from March 3, 2023, to March 16, 2023, at national level in Conakry and included 3 participants (Phase II, Table 1). The third phase was implemented from April 14, 2023, to April 22, 2023, and included 64 local participants from Phase I in the Temessadou-Djigbo and Koundou subprefectures (Phase III, Table 1). During this last phase, the preliminary results were presented to the participants involved in the first phase at the study sites in order to clarify, amend, and supplement the results presented on the basis of their previous statements and concerns, and to obtain their approval.

The thematic focus group and interview guides were pretested during 1 FGD and 2 IDIs with 12 participants, all from the decentralized (prefectural, regional), technical services and from the community (both the thematic and interview guides are available in supporting information S1 File and S2 File). After the pretest phase, a total of 16 FGDs and 21 IDIs were conducted. The topics discussed included (i) the alert and response protocols in place at the community, decentralized technical staff, and national stakeholder levels during previous Ebola, Marburg, and Lassa epidemics between 2014 and 2021, (ii) the constraints and factors for a successful response, (iii) the needs and expectations of the actors involved in the response systems, and (iv) the official alert response organization chart and how well actors (from the local to national levels) understood it. The number of participants per FGD varied between 6 and 12 people. The groups were homogeneous in terms of stakeholder categories (i.e., community members, community workers, local staff, prefectural staff, and national service staff), but not necessarily in terms of gender. This was because the presence of men in the same group did not prevent women from expressing their opinions. Based on knowledge of the local context, gender-related cultural sensitivity does not prevent individuals in mixed-gender group discussions from freely expressing their points of view when in a socioprofessional sphere. Environmental community workers were grouped with livestock community workers in FGDs because the surveillance of wildlife diseases in Guinea is carried out by the veterinary services [33]. FGDs and IDIs lasted between 40 and 90 minutes and were conducted in French, in the local language (Kissi), or in both French and Kissi by the principal investigator. The team also included an assistant to ensure that the interview ran smoothly, that stakeholder participation was effective, and that interventions remained consistent for the debriefing. A reporter was responsible for taking notes as well as making audio recordings and taking photographs. The team members had previously been trained in the use of participatory research approaches.

## Ethical framework

The study protocol received authorization from the National Ethics Committee for Health Research (CNERS) in Guinea in accordance with official acts No. 028/CNERS/22 of April 19, 2022 and No. 050/CNERS/23 of April 5, 2023. Approval from the local authorities was requested and obtained in each subprefecture of interest after the objectives of the study were explained to representatives. Before the interviews began (IDIs and FGDs), written consent forms were obtained for each participant or, in the case of FGDs, by a designated representative of the relevant stakeholder category.

Respondents were free to participate in the study without any obligation to answer all the questions. The interviews were recorded using a recording device, and notes and photographs were taken when relevant to the study and agreed by participants. The interviews were anonymized during the processing phase.

## Data processing and analysis

The interviews were transcribed in full and translated into French when necessary (with a Kissi–French translation provided by the principal investigator).

First, the reviewed transcripts, as well as notes from informal discussions, field notes, and diagram pictures, were imported into NVivo 2022 software (NVivo 14; formerly QSR International, now Lumivero). Second, the transcripts were classified according to interview type (FGDs and IDIs), stakeholder category data collection site (Gouécké town Temessadou Djigbo, Koundou, Temessadou M'Boké, Guéckédou, N'Zérékoré and Conakry), and decision-making level (local, decentralized, and national). The transcripts, pictures, and notes were classified and sorted and the relationships and trends in the data were examined. We conducted a thematic analysis where the main themes were identified using a deductive approach based on the study objectives. We then used an inductive approach to generate new themes and subthemes emerging from the FGDs and IDIs [34]. Following the iterative process of thematic analysis, the coding tree used to establish the themes was finalized after consensus between the main authors (the final coding tree is available in supporting information S3 File).

## Results

First, a mapping of the health stakeholders at the local (town/village, subprefectural), decentralized (prefectural, regional), and national levels was produced (Fig 2, S4 File). At the local level, the identified community members were local elected representatives, health matrons (respected women in the community who also serve as midwives), opinion leaders, healers, hunters, farmers, herders, woodcutters, teachers, and traditional village announcers. Community workers for the

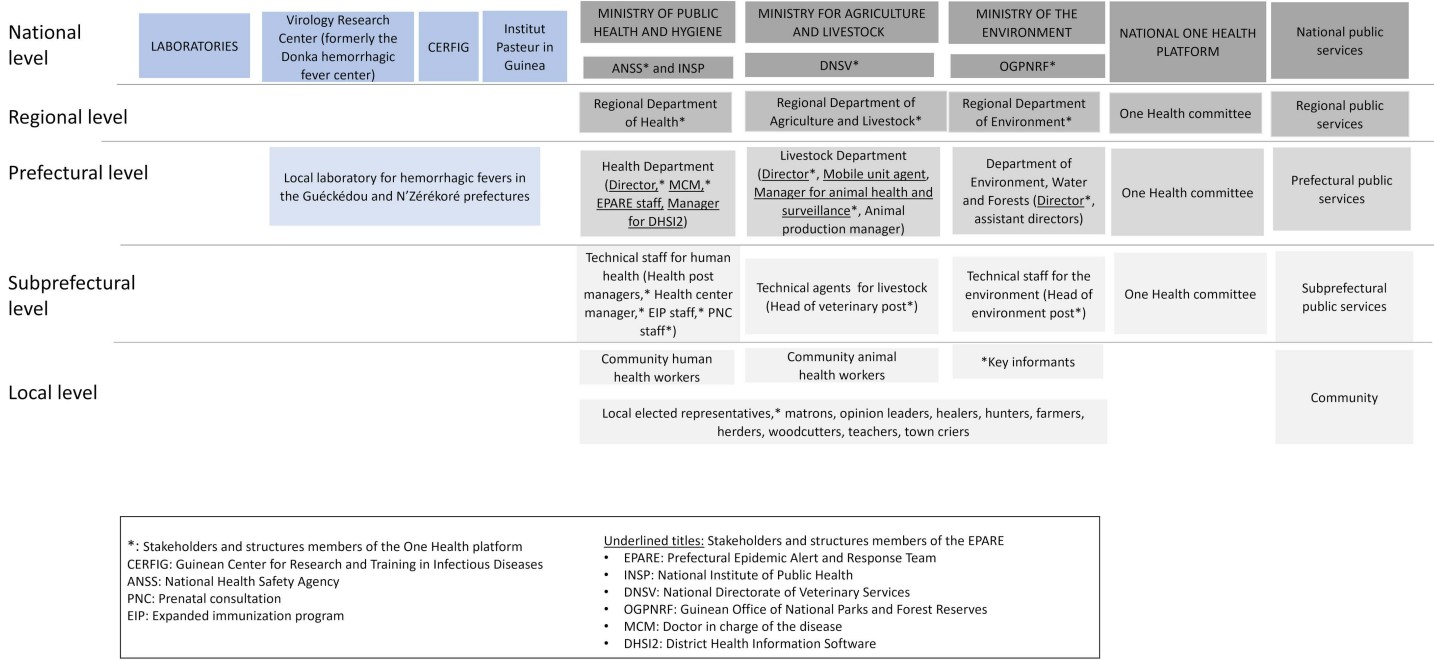

**Fig 2. Mapping of the stakeholders and organizations involved in outbreak response.**

human, animal, and environmental health sectors were also present at this local level, as well as decentralized technical service staff from the subprefectural administration for those same three sectors. At the prefectural level, we identified staff from the health, livestock and environment, and water and forest departments. Staff from the same sectors were identified at the regional level. Finally, at the national level, the main bodies in charge of disease surveillance and responses were the Guinean National Health Safety Agency (ANSS), the National Directorate of Veterinary Services (DNSV) and the Guinean Office of National Parks and Forest Reserves (OGPNRF), all members of the Guinean One Health Platform.

Among the 151 participants of the study and excluding the test phase, 124 (82.1%) were community workers and members of the community, who are considered frontline actors in case of alarming events; 24 (15.9%) were technical staff from local and decentralized services (subprefectural and prefectural services), and 3 (2.0%) were national actors. When broken down by gender, 30 (19.9%) of the study participants were women, compared with 121 (80.1%) men (Table 1).

The results presented below are organized by theme: (i) alarming events and alerts at local level, (ii) response flowcharts, (iii) obstacles, and (iv) levers in implementing response measures. They follow the path that information may take from the alarming event that triggers the signal through to the implementation of response measures. Any similar or different ideas mentioned by participants are included, and consider the stakeholder category, the data collection site, and the decision-making level. The main results of each theme are described in more detail and illustrated with anonymous quotes, chosen for their representativeness, appropriateness, and revealing quality. To highlight the observed patterns and respect as much as possible what was expressed in the discussions, participants' opinions and ideas are presented for each theme in order of the level of importance given by the participants to these topics (strong to weak level of consensus).

## Events and signal detection and reporting

Various signals described as alarming or worrying were mentioned by the participants. The signals were grouped into three categories: health signals (human and domestic or wild animal diseases or deaths), environmental signals (bushfires, floods, tornadoes), and sociopolitical signals (deaths and damage caused by strikes and demonstrations; Fig 3). According to the participants, each of these signals could lead to an event that could impact human and/or animal health, the environment, well-being, or food security. These signals alarmed communities because of their impact on daily life, including loss of life, loss of livestock and wildlife, reduced trade and commercial flows, and migration of citizens for fear of tougher response measures in the context of health crises. Regarding health signals, those identified by the majority of participants in the study led to alerts being sent to local and national surveillance authorities. These signals were based on cases that were observable through human and animal mortality, and most often through unusual clinical symptoms.

Various surveillance stakeholders at the local and national levels said that signals were reported by people who were recognized by local elected representatives, and who had voluntarily become involved as community workers, either in the human health, livestock, or environmental sectors. They also noted that signals were sometimes reported directly by community members themselves, without going through community workers. Considering the discussion on signals as a whole, each stakeholder category most often reported signals that concerned their area of activity. Signals concerning diseases were most often reported by the human and livestock health community workers, while the environmental community workers typically dealt with forestry problems, bushfires, and wildlife mortality.

## Outbreak response flowcharts

An official public health emergency management flowchart was developed and has been used in Guinea since the 2013–2014 Ebola virus disease epidemic crisis. Its aim is to improve alert management and response setup in the event of a an outbreak. This official flowchart was shared by the Prefectural Epidemic Alert and Response Team (EPARE) at the N'Zérékoré site (see diagram in Fig 4). This outbreak response flowchart, which is managed by the EPARE, specifies the steps to be followed based on the signal. This plan includes several transitional stages between the signal, the alert, and

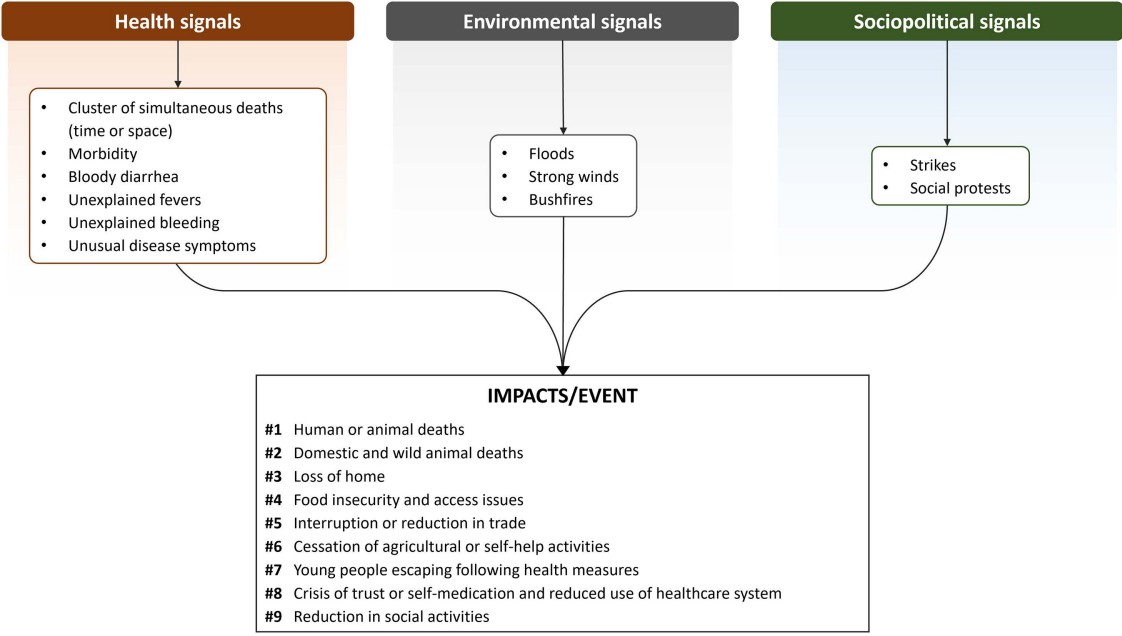

**Fig 3. Schematic representation of signals identified by local stakeholders and related negative impacts, built from focus discussion groups and in-depth interviews conducted in Guinea from 2022 to 2023, ranked from 1 (most impactful) to 9 (least impactful).**

the implementation of the response to a health event, corresponding to the steps required to verify and validate the health event by various entities, until the corresponding response action is implemented. Participants at the local level were asked about their understanding of this flowchart. The technical service employees who were interviewed said this flowchart was difficult to understand and use, as it contained unconventional abbreviations and acronyms and it lacked a color legend and meaning. The official chart identified the emergency operations center (EOC) as the main body responsible for response activities in the Guinean National Health Safety Agency and did not clearly indicate the role and tasks of the other entities involved in implementing the response, including local actors. The level of activation of the response mechanisms in the flowchart depended on the alert threshold and epidemiological threshold for activating the response at the level of the EOC- and the EPARE. However, the threshold, timeframe for these actions, and people responsible for them at the local level were not specified.

Although community workers were unable to fully understand the official flowchart, they still took action based on what they did understand and on their past experiences with epidemic responses. The research team recreated the flowchart as community workers understood it based on information provided during IDIs and FGDs (Fig 5). During Phase III, this reconstructed flowchart was then amended and validated by the same staff at local level. The perceived outbreak response flowchart describes the roles and activities carried out by local staff as part of the response within the communities. This flowchart comprises several stages, from the signal to the response implementation at local level by the community workers and local staff and then at the national level. The official and perceived outbreak response flowcharts differ in the description and distribution of tasks among all stakeholders. In our reconstructed flowchart, the alert starts with community members or community workers. This signal is first verified by decentralized technical staff at the subprefectural level, under the EPARE's supervision. If the signal is positive, samples are sent to the prefectural laboratories and/or the local VHF laboratory (Fig 2 and Fig 5). Then, in terms of response, each entity (community workers, technical services) and staff from national services respond in different ways if the laboratory results are positive or negative. If the result is negative, all stakeholders inform the other staff, with community workers and decentralized technical staff informing community members. If the result is positive,

**Fig 4. Schematized public health emergency management flowchart (Source: EPARE- N'Zérékoré).** The authors determined the meanings of the acronyms based on interviews with the study participants. PRO: procedure to follow; HCM: health center manager; CTEpi: epidemiological treatment center; EOC: emergency operations center; PHD: prefectural health department; EPARE: Prefectural Epidemic Alert and Response Team; SITREP: situation report, Lab: laboratory; MPE: unable to determine; IMS: incident management system; OH: One Health; PAC: patient care; PL: unable to determine; CPP: unable to determine; SOPs: standard operating procedures.

awareness-raising actions, crisis meetings, training, and direct responses such as contact follow-up, patient transfer, and sanitary zoning are organized by community workers and subprefectural technical staff under the supervision of EPARE. This flowchart shows that community workers play a key role in implementing local response measures. They alerted neighbors and local elected officials. They also raised awareness among the population in order to promote and encourage basic self-protection and control measures. For example, in the event of a suspected zoonosis, community members were advised to self-report any suspicions, adopt good handwashing practices, practice social distancing, and prohibit handling and eating animals considered to be pathogen reservoirs as well as any other dead animals.

The IDIs carried out with the national stakeholders (livestock services, national One Health and human health platform) showed that the response was generally implemented following verification of the signal at the local level by the decentralized technical stakeholders and confirmation of the samples by laboratories.

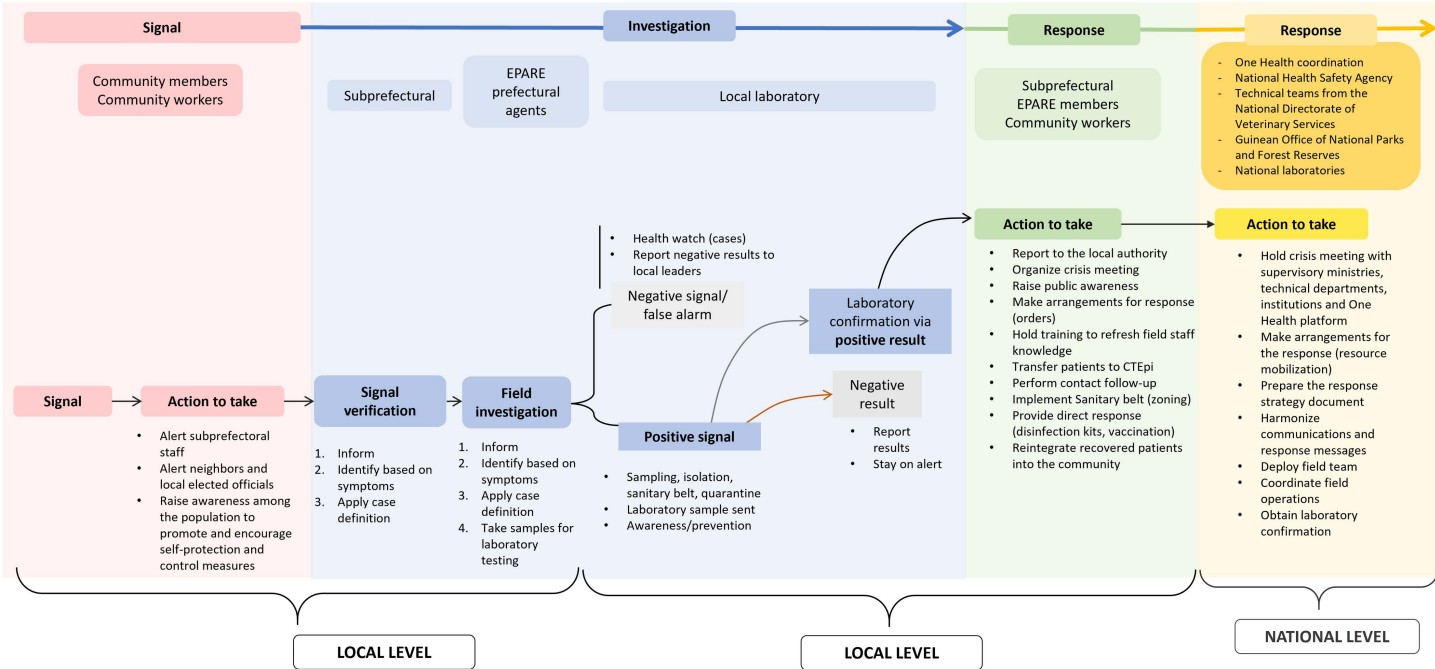

**Fig 5. Re-creation of the perceived response in case of an infectious disease signal.** EPARE: Prefectural Epidemic Alert and Response Team.

## Obstacles to and levers for implementing effective and sustainable response measures

Six main obstacle categories were identified from the interviews: (i) lack of logistical and financial resources, (ii) lack of technical skills in surveillance and early detection, (iii) lack of legitimacy, (iv) lack of coordination, (v) a large number of actors from health institutions in the villages, which leads to (vi) communities' weariness with regard to multiple and uncoordinated actions. According to one prefectural livestock officer, an organizational obstacle was the disparity in knowledge between the different sectorial players. He noted that human health stakeholders benefited from more training and financial resources than other sectors. Similarly, local livestock and environmental staff stated that the Ministry of Health had greater financial, logistical, and technical capacities than the Ministries of Livestock and Environment. This would lead to a possible reduced involvement and motivation in these two sectors due to insufficient financial means required for the regular monitoring of post-impact activities and ongoing awareness-raising.

Another category participants mentioned was the lack of technical skills on surveillance and response to zoonotic disease emergences. According to staff from the national livestock services, the Ministry of Health and Public Hygiene and the Ministry of Livestock (National Veterinary Services Department) staff have basic skills related to their profession, but there is a lack of initial training in surveillance and response when they take up a new position. Similarly, in the Ministry of Environment, although a wildlife disease surveillance system is currently being developed, there is still a shortage of competent wildlife disease surveillance officers. This issue was reported by field workers as an obstacle to the early detection of wildlife diseases and the effective flow of health information.

In addition to these issues, the role of the community workers is not always formalized, which makes their work difficult and can lead to a lack of legitimacy. This lack of legitimacy highlighted by animal and environmental health community workers. They pointed out that they were not identified as community workers by villagers, unlike human health workers, whose role was identified by their uniforms and badges.

There can also be inconsistencies and duplication of response activities between international institutions, which is seen as an obstacle by the communities of Temessadou and M'Boké. This lack of coordination between local authorities, research institutes, and international institutions during outbreaks was reported by the participants. They highlighted a need for coordination during outbreaks, especially for the benefit of communities. For example, during Ebola or Marburg outbreaks, some of the research and response activities, such as bat sampling in caves and human sampling, were duplicated within villages, which was intrusive for communities and provoked negative reactions to the presence of repeat teams in villages. One community stakeholder reported: *"We're tired of them coming to ask us every day about the Marburg disease. They came here after three months of investigation to declare the end of the disease and told us that we were free to move. Then, we were sent to Guéckédou to the rural radio station to talk about this disease and the means of prevention and control that the village accepted; so a few months after the disease was declared defeated, I saw other teams here, people started to be afraid and to spread word to oppose and chase them away. It's the same things – it's worrying for us and we need to express our dissatisfaction with certain things"* (translated from Kissi, FGD/community/ Temessadou M'Boké).

Multiple and uncoordinated actions make communities weary, which can cause them to reject new actions. Local people in Temessadou M'Boké perceived the investigation, responses, and research actions as invasive within their territory, in turn hindering the implementation and acceptance of response measures. One participant noted that "Institutions or technical partners sometimes make false promises in their communication, leading to subsequent claims by communities" (translated from Kissi, FGD/community/Temessadou M'Boké).

From these obstacles, three main categories of needs were identified: (i) financial, (ii) logistical, and (iii) training.

During the FGDs, human health, livestock and environment community workers, and subprefectural staff mentioned a need for lump sums to be granted. In addition to formalizing their status, this would represent a significant boost to their involvement in surveillance activities and response measure implementation. Participants also listed other key needs such as financial resources to run health facilities (village watch committees, village One Health platforms) and pay for telephone credits and fuel costs to facilitate travel to certain remote and isolated areas. In a similar vein, personal protective equipment such as gloves, boots, and coats for the rainy season were identified as needs to implement responses properly. Participants also requested means of transport, awareness-raising fact sheets, and picture boxes to help understand the diseases and associated clinical signs. All the FGDs and IDIs at all levels and collection sites emphasized the need to train local and decentralized technical staff at the prefectural level on topics such as wildlife disease surveillance and sampling procedures. For community workers (human health, animal, environment), training in disease recognition, prevention and control methods, and risk awareness and communication was mentioned. The same actors recognized that reviewing and refreshing their knowledge on what was learned during previous training courses was essential. The revitalization or setup of health monitoring facilities and One Health platforms was also mentioned as a need.

## Discussion

Lessons learned from several outbreaks show that local community and local stakeholder engagement is crucial for effective outbreak responses [15,16,18]. Our qualitative study was an opportunity to give a voice to these local actors to better understand the health signals that alert them to a problem and describe the outbreak response measures that are implemented at local level. Our study also helped identify the obstacles and levers for implementing community-based response plans adapted to the sociocultural context and needs of local stakeholders in Guinea.

### Limits

A few biases should be mentioned regarding our study. First, memory bias – the way in which a person remembers an event – can affect results because respondents may forget details or have systematic errors in thinking [35]. In our study, participants only remembered recent events, such as the 2021 outbreaks of Marburg and Ebola, or could not accurately

describe events that occurred several years ago, such as 2013–2016 Ebola outbreak. However, we considered that memory bias was minimized by the large number and wide range of stakeholders interviewed, data triangulation, and the amendment and validation of the findings by participants [36]. Some categories of stakeholders, mostly national representatives, were underrepresented because of their unavailability and busy schedules. This resulted in a perceived outbreak response flowchart that only takes into account local stakeholders' points of view. There is also a social bias due to the low participation rate of women in the FGDs. This constitutes a source of bias because women had been identified as the preferred interlocutors for family health issues in a previous study [20]. The low participation rate of women in the study can be explained by the fewer number of women working in technical services, such as veterinary services, at national level. Thus, our results need to be taken with caution as the management of alert and response mechanisms might be different if more women were included in our study as informal and nontechnical actors involved in alert and response management. We were able to partially offset this bias by including community health matrons and health centers in the FGDs, as they are extremely familiar with the health problems of women and children in the villages. Finally, our study was conducted in Guinée Forestière where the main outbreaks occurred. It could be interesting to extend this survey to other non-peri-urban sites and to regions of Guinea that have not experienced major epidemic crises. Discussions could be held with communities on signals (whether health-related or environmental) that appear relevant to them. Guinea is a vast country with varied local ecological and socio-cultural characteristics that need to be addressed. Having a wider diversity of stakeholders would be helpful in adapting alert and response systems to public health emergencies at national level with a view to curbing future epidemics. Our survey should also be disseminated to policymakers to improve rapid respond measures.

## Main findings

Local stakeholders identified several alarming signals related to health, environmental, and sociopolitical issues. These signals led to events that could impact human and animal health, environment, well-being, or food security. Regarding health signals, we described the official outbreak response flowchart and a second flowchart as perceived by local stakeholders. The official flowchart was developed at the national level based on a top-down approach, whereas we developed the second one as part of this study with a bottom-up approach in conjunction with local stakeholders.

The official flowchart seems to be well adapted to national and decentralized stakeholders who need an overview of the sequence of actions and actors involved in responding to an outbreak. However, it contains many acronyms and technical terms that are not explained within the document itself (e.g., alert threshold), making it difficult for local stakeholders to understand. Both flowcharts start with a signal reported by community members or community workers. However, the official flowchart seems narrowly focused on human health signals, whereas our flowchart can be used by stakeholders in all three sectors (health, environment, sociopolitical). Our work acknowledges that community workers from all three of these sectors raise alerts relating to signals specific to their specific sectors.

Timeliness, such as early signal detection and early investigation, has been highlighted as a key factor in an effective outbreak response [37–39]. Clarity around all stakeholders' roles and responsibilities along with good coordination among these various stakeholders at different levels is also vital [18,37]. Lack of knowledge about to whom to report during the Sudan virus disease outbreak in 2022 in Uganda was, for example, considered a weakness for community-based surveillance and led to delays in disease detection [18]. We can draw from our study findings to suggest concrete health signals and a response framework developed by community members and community workers. This type of bottom-up approach would make local stakeholders more fully aware of their roles and responsibilities in surveillance and outbreak response. Previous research has demonstrated the essential role of community-based surveillance and response systems in early detection, investigation, and response with regard to outbreaks such as Sudan disease virus in Uganda and measles and monkeypox in Cameroon [18,40]. These later findings highlight the importance of developing surveillance, investigation, and response flowcharts that are understood by community members and community workers. Creating these flowcharts

 

with local stakeholders as we have done could be an effective way of enabling them to take ownership, and consequently better integrate their roles and responsibilities in responding to health emergencies. Additionally, lack of all-cause mortality surveillance was identified as a gap that could potentially contribute to delayed outbreak detection [18]. Indeed, mortality is often associated to clinical signs in surveillance case definition. Our results show that from a community point of view, signals leading to adverse events such as mortality and morbidity, whether infectious or environmental, are just as important. This opens the door to the implementation of an integrated surveillance and response system that would consider health events and natural disasters as signals that could have an impact on public health. According to the various statements made by local and national staff about the success of the response teams, some delays in implementing response measures were still reported. For example, in some areas, citizens opposed vaccination tents being set up during the 2021 Ebola outbreak. The same findings or similar results were observed during the previous Ebola epidemics of 2018 in the Democratic Republic of the Congo [16]. Communities in Guinea were subjected to imposed intervention measures, unlike in Liberia and the Democratic Republic of the Congo, where community-based intervention strategies were implemented and better accepted [15,16]. In addition, there may be inconsistencies and duplication of response activities between international institutions, which is considered an obstacle by the communities of Temessadou M'Boké. Participants pointed out poor coordination as a key issue and stressed the need for enhanced coordination during epidemics, especially in ways that benefit communities.

### Using qualitative research to discuss complex health issues and identify levers

Measures taken based on a top-down approach, without considering the constraints and obstacles faced by field stakeholders, are often not accepted by communities and so are ineffective and unsustainable [41,42]. The reasons for this lack of community acceptability involve complex processes. Qualitative and participatory approaches have been found to be effective for discussing complex health issues by considering individual characteristics and societal influence on health determinants and developing acceptable health management in Uganda [18]. Indeed, using qualitative and participatory approaches in our study enabled us to understand the local knowledge of populations and identify key upstream actors and other hidden or hard-to-reach stakeholders [27,28]. Finally, including different categories of stakeholders in our study allowed us to collect a range of viewpoints.

### Recommendations for overcoming obstacles

Several recommendations can be made from this perspective. First, while the official outbreak response flowchart appears to be well-suited for the national, regional, and prefectural levels, a simplified version could enhance the engagement of local stakeholders in outbreak responses. The revised flowchart we developed, based on interviews with local stakeholders and their validation, could serve as a useful starting point to co-construct a more suitable and integrated response system.

Second, throughout our study, participants highlighted the financial, logistical, and training disparities between the human health sectors on one hand and the animal and environmental health sectors on the other. These disparities may lead to weaker involvement and motivation of animal and environmental health stakeholders. Similar findings were observed in the same contexts during the prioritization of diseases to monitor at national level in Guinea [20,43] and during the investigation of the Ebola virus alert system management tool in South Sudan [44]. To address these disparities, sharing local resources – e.g., through the local One Health platform – and taking a long-term approach to the training and logistical needs of each sector would help equalize and bridge the gaps between the three sectors.

The lack of legitimacy afforded to community workers and insufficient financial resources devoted to them was also highlighted as a problem. Indeed, during all the FGDs and IDIs, community workers and local staff mentioned the lack of equipment and financial incentives needed to boost and improve alert-raising efforts by community workers. These unmet needs could lead to loss of motivation on the part of these players, who are essential for relaying information from

the source of disease emergence events. Community acceptance of community workers and motivation of these actors have been identified as drivers of success for community-based surveillance [45]. As already observed in Ghana, Sierra Leone, and South Sudan, providing training opportunities, equipment and financial incentives such has telephone credits, and coverage of travel costs are key ways to ensure success in raising and reporting alerts [45,46]. Finally, community members noted that institutions make promises during outbreaks that they do not deliver or implement. These unfulfilled promises lead to frustration and a loss of trust between community members and the institutions or government in charge of responding to epidemics, as observed in Liberia and Sierra Leone [47,48].

The overall resilience of healthcare systems in countries depends in part on the ability of those countries to rapidly detect and respond to outbreaks. Community-based response systems may enhance this rapid response, provided that the key concepts of epidemiological disease surveillance and community social dynamics are fully taken into account.

## Supporting information

**S1 File. Thematic guide for Focus Group Discussions.**
(DOCX)

**S2 File. Thematic guide for In-Depth Interviews.**
(DOCX)

**S3 File. Coding tree.**
(DOCX)

**S4 File. Description of stakeholders, organizations, and roles.**
(DOCX)

## Acknowledgments

We would like to extend our sincere gratitude to the Guinean National Department of Veterinary Services (DNSV) and in particular Dr. Leonce Zogbelemou, prefectural director of livestock in Macenta, and Dr. Momory Leno, head of the veterinary center in Temessadou, for their assistance in the field. We would also like to thank Celia Lacomme, research engineer at CIRAD, for her logistical and organizational support. Our deepest appreciation also goes to all the prefectural and subprefectural authorities for their understanding and welcome, and to all study participants and local authorities for their participation and the trust they placed in us. The authors are also grateful to Teri Jones-Villeneuve for the English editing.

## Author contributions

**Conceptualization:** Marie-Marie Olive, Saa André Tolno, Severine Thys, Alpha Kabinet Keita, Véronique Chevalier.

**Data curation:** Saa André Tolno.

**Formal analysis:** Marie-Marie Olive, Saa André Tolno, Severine Thys, Chloé Bâtie.

**Investigation:** Saa André Tolno, Maxime Tesch.

**Methodology:** Marie-Marie Olive, Saa André Tolno, Severine Thys, Alpha Kabinet Keita, Maxime Tesch, Véronique Chevalier.

**Project administration:** Marie-Marie Olive.

**Supervision:** Marie-Marie Olive, Severine Thys, Alpha Kabinet Keita, Véronique Chevalier.

**Validation:** Marie-Marie Olive, Saa André Tolno, Severine Thys, Alpha Kabinet Keita, Véronique Chevalier.

**Visualization:** Marie-Marie Olive, Saa André Tolno, Maxime Tesch.

**Writing – original draft:** Saa André Tolno.

**Writing – review & editing:** Marie-Marie Olive, Severine Thys, Alpha Kabinet Keita, Maxime Tesch, Chloé Bâtie, Véronique Chevalier.

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
