## [Decision Letter · Decision Letter 0]

5 May 2025

Dear Dr. Olive,

Thank you for submitting your manuscript to PLOS ONE. After careful consideration, we feel that it has merit but does not fully meet PLOS ONE’s publication criteria as it currently stands. Therefore, we invite you to submit a revised version of the manuscript that addresses the points raised during the review process.

We look forward to receiving your revised manuscript.

Kind regards,

Ted Loch-Temzelides

Academic Editor

PLOS ONE

Journal Requirements:

3. In the online submission form, you indicated that [The anonymous transcript of the interviews would be available in a dataverse uppon request to the corresponding author.].

Additional Editor Comments:

This study investigates the role of local workers and responders in outbreak response interventions, focusing in the forest area of Guinea, an area vulnerable to zoonotic disease emergence. The study qualitatively explores how communities in Guinea react to hemorrhagic fever outbreaks and considers how local input can reinforce existing health systems and improve epidemic preparedness. It highlights that local populations can detect health, environmental, and social signals of potential outbreaks. Such observations can lead to more grounded and effective responses. The research also identifies some factors that impede community-based responses, including limited resources, lack of appreciation for community health workers, and poor coordination with official health systems. The study demonstrates the value of integrating local perspectives into outbreak management, supporting early detection, better communication, and stronger stakeholder involvement.

All three referees, who are experts in the field, indicate that an appropriately revised version of the paper could make a good contribution to PLOS One. Reviewers 1 and 3 make several comments that would clarify the analysis and improve exposition. As you can see, there is a significant overlap in their comments. Notably, the paper would clearly benefit from edits from an English language editor. This is necessary and it is not just about cosmetics. As it stands, certain parts of the paper are hard to read. In addition, some of the main concepts need to be more clearly defined, and the references cited need to be clarified. Reviewer 3 points to several such parts and discrepancies in the paper.

I am looking forward to reading the revised version of the paper that addresses in detail each point made by the referees.

Reviewers' comments:

Reviewer's Responses to Questions

**Comments to the Author**

1. Is the manuscript technically sound, and do the data support the conclusions?

Reviewer #1: Yes

Reviewer #2: Yes

Reviewer #3: Yes

2. Has the statistical analysis been performed appropriately and rigorously?

Reviewer #1: N/A

Reviewer #2: Yes

Reviewer #3: N/A

3. Have the authors made all data underlying the findings in their manuscript fully available?

Reviewer #1: Yes

Reviewer #2: No

Reviewer #3: Yes

4. Is the manuscript presented in an intelligible fashion and written in standard English?

Reviewer #1: Yes

Reviewer #2: Yes

Reviewer #3: No

Reviewer #1: The article underscores the importance of community partnership in addressing complex, "wicked" problems, highlighting that collaborative, bottom-up approaches tend to yield more sustainable and effective outcomes than top-down strategies. I commend the authors for their thoughtful and methodical efforts to foster genuine community engagement and secure local buy-in. It is striking, however, that this lesson must be continually relearned in public health and policy work.

The authors adhered to established ethical protocols and conducted comprehensive engagement with local communities across key sectors. Their aim was to understand community responses and needs during critical outbreaks of Ebola Virus Disease (EVD) and other viral hemorrhagic fevers. Through qualitative methods and an emphasis on participatory research at every stage, the study successfully identified barriers to effective public health responses. This approach enabled the authors to develop contextually appropriate guidelines that align with the realities of decentralized technical services.

Reviewer #2: The manuscript makes a very important contribution to the development of community-based response systems. This qualitative participatory study investigated rapid community-based responses to haemorrhagic fever outbreaks in Guinea, emphasizing the involvement of local stakeholders to enhance effectiveness and sustainability. Local communities recognized various health, environmental, and socio-political signals prompting outbreak alerts. Researchers identified significant barriers, including limited logistical and financial resources, insufficient legitimacy of community workers, and poor coordination. Co-developed response flowcharts reflecting local stakeholders' experiences and perceptions provided practical guidance, fostering clearer understanding and commitment to response measures. The findings underscore the importance of integrated, bottom-up community-based systems for early detection, improved stakeholder engagement, and effective epidemic management.

I started to do some corrections, but left it at some point, realizing that the article requires professional language editing by native English speaker.

Hence, I recommend a full language editing.

- Line 41: in the response

- Line 42: measures instead of measure, were consequently less understood and failed to engage local stakeholders

- Line 50: environmnetal, and

- Line 51: “local stakeholders“ with the

- Line 82: related-health signals is odd

- Line 82-83: However, the coordination of …. is insufficient,…

- Line 86 : sentence is odd

- Line 87: Either “Transmissions” or “The transmission”

Reviewer #3: The overall research highlights the importance of engagement and co-design of outbreak response interventions starting with local workers or responders. While not a new topic it provides insight to the Forest part of Guinea, a known zone at-risk of zoonotic disease emergence and can strengthen current system. The qualitative approach leading to bottom-up recommendations that would complement top-down flowcharts and decision-making is well described and brings a great opportunity to engage multiple stakeholders that are critical for detection in any event.

Overall the article would benefit from re-structuring its content and be more to the point, providing more insights as of what resulted from the interviews and the focus groups. It would be useful to provide the analytic grid that were used for both in the supplementary material. More importantly, the article needs in-depth proof-read in English as currently it is difficult to read. Some concepts or terms would also benefit from being clearly defined (see details below).

Literature search strategy is not mentioned and it feels that references are more quoted for convenience rather than following systematic search. As an example reference 5, Magassouba et al. is quoted on line 69-70 indicating that “recurrent cases of Lassa fever have been regularly reported in all regions of the country since 2017”, when the reference does actually not support this claim. Some references are included several times in the reference list (see ref 3, 4 and 5 that are respectively referred also in ref 29, 28 and 30)

Introduction

• Authors should consider providing a definition of what they include in VHF. It seems that only filovirus diseases (Ebola virus disease & Marburg virus disease) and Lassa fever are included. It may worth specifying it. Burden of Lassa fever was well described in the 90s, but it is not clear how authors can back-up their claim that Lassa fever “has been in all regions of the country since 2017” when the referenced article only mentions two confirmed cases in Macenta and Kissidougou.

• Authors may consider to add a brief description of the different roles/levels involved in disease surveillance in the Forest Guinea so that it helps contextualize.

Material and methods

• It would be good to explain the sampling size, how the number of IDI & FGD was selected. What was the target and how many actually happened. Currently it reads like the number was more based on the convenient sample.

• It would be good to provide, possibly in supplementary materials what are the functions of the different groups interviewed (what does their functions actually involve on a daily basis).

• On the description of the IDI and FDG :

o Design of table 1 would benefit from redesign to make it easier to read. At least repeat heard row on each page of the table.

o Why are there IDIs with several interviewees cf. IDI 2, 3,4, 5 and 6? Was the same guide used for IDI/FDG or was it different thematic. It is not really clear when looking at the table.

o As it reads in table 1, 93 participants were interviewed in Phase 1, 3 in Phase 2 and 64 in phase 3. You may want to double check the difference between number participants for phase 1 mentioned on line 183 (91) to see if there was not a typo in the table.

o In Table 1, data are presented for 11 IDI and FDG. It is explained in lines 199-201 that 2 IDI and 2 FDG were used to validate the tool and that it included xx participants. Table 1 reaches a total of 160 participants when “only” 158 are mentioned, and if we removed the 12 participants from the 2 IDI and FDG, then it should only be 146 participants in table 1. Please review the discrepancy.

o Please explain what is the difference between phase 1 and 2? Was this because interviewees interviewed in phase 2 were not available in phase 2?

o One line 207 it is mentioned that FDG had 6 to 12 participants, please explain why several FDG had a total of more participants cf. FDG 1, 3, 8, 9 etc. – or was it several FDG on the same topic and/or with same target group?

• Would the authors consider providing the interview guides (IDI and FDG) in supplementary material?

• Lines 210-2012 : please consider rephrasing sentence as it is not clear in current format and that later in the limitation there is some discussions as of why less women were included and how it could impact perception. It is not clear in these lines whether or not authors consider that gender in-balance may be a limitation or not.

• You may want to consider providing a definition to “central”, “deconcentrated” and “local”. Also in the document, “deconcentrated” and “decentralized” seem to be used interchangeably cf. lines 253, 263, 334-335– these terms are not synonyms or if used in a similar way this should be explained and defined.

Results

• Lines 261-268 : it is mentioned that results will be presented through 5 themes. Last theme “perception of control measures” should include a headline similarly to the other themes and be developed on the result section – or be removed from text. It currently does not appear.

• It would be good to describe the weighting of the themes that came out from the IDI and FDG. How were they classified and weighted to come out with the one selected. It’d be good as well to quantify (as possible) how they were perceived by category of interviewees (for ex. 80% of x group reported issues with access to resources for instance).

• Fig. 5 – it’s great to see the representation of the flowchart as perceived by local stakeholder. I was wondering if it was at all possible to include immediate response that is described in text (lines 345-350) at local level. In current format, it seems from fig. 5 that no response activities happen at local level which it contrary to what is mentioned in text. It’d be good to visualize this as well.

• Legitimacy or rather perception of being legitimate is only briefly mentioned as an obstacle, you may want to consider elaborating slightly more on this.

• Lines 348-399 : for the first time in this article, “international” oragnizations are mentioned. It is a really interesting finding and in particular to note that it does not appear in any of the flowcharts. It’d be good to further discuss this in the discussion session as it seems to be a challenge and emphasize the lack of coordination as perceive by local stakeholders.

• Line 403-406 : the extract of the verbatim seems to point out towards an issue that is not really mention throughout the article, that is promises that are made to communities by a institutions but not delivered or implemented. This is something observed in many outbreaks and leads to trust issues and has an impact on implementation of interventions. This is something that you also may want to explore further in the discussion section.

Discussion

• You may want to restructure slightly the discussion section, maybe moving the limits part first rather than in the middle of the section.

• You may want to move lines 451-457 after you complete discussion on the alert / signal detection and reporting to follow the flow of action (that is moving it after line 479).

• You may want to add reference to back up statement at line 455.

• Lines 474-479 : this part on mortality surveillance is not really clear to me. You may want to clarify your statement and specifying if mortality surveillance is something that is already in place or that should be improved.

• Lines 510-515 : you may also want to state that similar approach may need to be reiterated in other Prefectures / Region focusing on relevant diseases so that flowchart is somewhat similar but adapted to local level. It would be good to discuss this, especially in a large country with different ecologies therefore with different risks in different habitat.

**Do you want your identity to be public for this peer review?** For information about this choice, including consent withdrawal, please see our Privacy Policy

Reviewer #1: **Yes: ** Patricia A Omidian

Reviewer #2: No

Reviewer #3: **Yes: ** Anaïs Legand

---

## [Author Response · Author response to Decision Letter 1]

7 Jul 2025

Marie-Marie Olive,

Researcher at ASTRE Unit, Cirad

marie-marie.olive@cirad.fr

Montpellier, June 24, 2025,

To the editorial team of Plos One journal,

Dear Editor,

We would like to thank you for considering our manuscript [PONE-D-25-09977].

We also would like to gratefully acknowledge the three reviewers for their positive feedbacks and encouragements. We provide below responses to each point raised by reviewers.

Finally, the manuscript, figures, and supporting information have been reviewed by a native English-speaking copyeditor.

We hope that you will further consider this article for publication in Plos One.

Yours Sincerely,

Marie-Marie Olive

Reviewer #1:

The article underscores the importance of community partnership in addressing complex, "wicked" problems, highlighting that collaborative, bottom-up approaches tend to yield more sustainable and effective outcomes than top-down strategies. I commend the authors for their thoughtful and methodical efforts to foster genuine community engagement and secure local buy-in. It is striking, however, that this lesson must be continually relearned in public health and policy work.

The authors adhered to established ethical protocols and conducted comprehensive engagement with local communities across key sectors. Their aim was to understand community responses and needs during critical outbreaks of Ebola Virus Disease (EVD) and other viral hemorrhagic fevers. Through qualitative methods and an emphasis on participatory research at every stage, the study successfully identified barriers to effective public health responses. This approach enabled the authors to develop contextually appropriate guidelines that align with the realities of decentralized technical services.

Authors’ response: The authors would like to gratefully acknowledge reviewer 1 for this positive feedback and encouragement.

Reviewer #2:

The manuscript makes a very important contribution to the development of community-based response systems. This qualitative participatory study investigated rapid community-based responses to haemorrhagic fever outbreaks in Guinea, emphasizing the involvement of local stakeholders to enhance effectiveness and sustainability. Local communities recognized various health, environmental, and socio-political signals prompting outbreak alerts. Researchers identified significant barriers, including limited logistical and financial resources, insufficient legitimacy of community workers, and poor coordination. Co-developed response flowcharts reflecting local stakeholders' experiences and perceptions provided practical guidance, fostering clearer understanding and commitment to response measures. The findings underscore the importance of integrated, bottom-up community-based systems for early detection, improved stakeholder engagement, and effective epidemic management.

I started to do some corrections, but left it at some point, realizing that the article requires professional language editing by native English speaker.

Hence, I recommend a full language editing.

Authors’ response: The authors would like to express their gratitude to reviewer 2 for these positive comments and encouraging feedback. The reviewer’s suggested edits (see below) have been made and the manuscript, figures, and supporting information have been reviewed by a native English-speaking copyeditor.

- Line 41: in the response Edit made

- Line 42: measures instead of measure, were consequently less understood and failed to engage local stakeholders Edit made

- Line 50: environmnetal, and Edit made

- Line 51: „local stakeholders“ with the Edit made

- Line 82: related-health signals is odd Edit made

- Line 82-83: However, the coordination of …. is insufficient,… Edit made

- Line 86 : sentence is odd The sentence has been rephrase

- Line 87: Either “Transmissions” or “The transmission” Edit made

Reviewer #3:

1. The overall research highlights the importance of engagement and co-design of outbreak response interventions starting with local workers or responders. While not a new topic it provides insight to the Forest part of Guinea, a known zone at-risk of zoonotic disease emergence and can strengthen current system. The qualitative approach leading to bottom-up recommendations that would complement top-down flowcharts and decision-making is well described and brings a great opportunity to engage multiple stakeholders that are critical for detection in any event.

Authors’ response: The authors would like to thank reviewer 3 for this positive feedback.

2. Overall the article would benefit from re-structuring its content and be more to the point, providing more insights as of what resulted from the interviews and the focus groups. It would be useful to provide the analytic grid that were used for both in the supplementary material.

Authors’ response: Thank you for asking for more insights with regard to the analytical process. Since we began with a deductive approach, our analytic grid first reflected the thematic guides we used to conduct the focus group discussions and the interview guides we used during semi-structured interviews. Within the iterative process and new themes that emerged from inductive coding, the data were finally coded and grouped into similar categories – this is what we named “coding tree” in the manuscript. This coding tree corresponds to the analytic grid, and we have now added the final coding tree to supplementary material S3.

3. More importantly, the article needs in-depth proof-read in English as currently it is difficult to read. Some concepts or terms would also benefit from being clearly defined (see details below).

Authors’ response: Thank you for this comment. The manuscript, figures, and supporting information have been entirely reviewed by a native English-speaking copyeditor.

4. Introduction : Literature search strategy is not mentioned and it feels that references are more quoted for convenience rather than following systematic search. As an example reference 5, Magassouba et al. is quoted on line 69-70 indicating that “recurrent cases of Lassa fever have been regularly reported in all regions of the country since 2017”, when the reference does actually not support this claim.

Authors’ response: Thank you for this comment. We have modify the introduction accordingly and added the following reference (highlighted in yellow lines 87 to 88 track changes version and lines 73 to 74 unmarked version). : Millimouno TM, Meessen B, van de Put W, Garcia M, Camara BS, Christou A, et al. How has Guinea Learnt from the Response to Outbreaks? A Learning Health System Analysis. BMJ Global Health. 2023;8:e010996. doi: 10.1136/bmjgh-2022-010996.

5. Introduction : Some references are included several times in the reference list (see ref 3, 4 and 5 that are respectively referred also in ref 29, 28 and 30)

Author’s response: Thank you for your vigilance. We have checked the references and removed any duplicates.

6. Introduction : Authors should consider providing a definition of what they include in VHF. It seems that only filovirus diseases (Ebola virus disease & Marburg virus disease) and Lassa fever are included. It may worth specifying it.

Author’s response: Thank you for your suggestion. We have clarified this in the first sentence of the introduction (highlighted in yellow line 80 track changes version and line 66 unmarked version).

7. Introduction : Burden of Lassa fever was well described in the 90s, but it is not clear how authors can back-up their claim that Lassa fever “has been in all regions of the country since 2017” when the referenced article only mentions two confirmed cases in Macenta and Kissidougou.

Author’s response: Thank you for your vigilance. We have modified the introduction accordingly: “Cases of Lassa fever have also been reported in recent years in Forest Guinea [5, 6]”. [5, 6].” (highlighted in yellow lines 86 to 87 track changes version and lines 73 to 74 unmarked version). We also added the following reference: Millimouno TM, Meessen B, van de Put W, Garcia M, Camara BS, Christou A, et al. How has Guinea Learnt from the Response to Outbreaks? A Learning Health System Analysis. BMJ Global Health. 2023;8:e010996. doi: 10.1136/bmjgh-2022-010996.

8. Introduction : Authors may consider to add a brief description of the different roles/levels involved in disease surveillance in the Forest Guinea so that it helps contextualize.

Author’s response: Thank you for this suggestion. Since we describe the surveillance and response stakeholders in detail in the Materials and Methods section, in Figure 2, and in the supporting information file S4, we would prefer avoid repeating this information in the introduction.

9. Material and methods : It would be good to explain the sampling size, how the number of IDI & FGD was selected. What was the target and how many actually happened. Currently it reads like the number was more based on the convenient sample.

Author’s response: Indeed, there was no target sampling size for the IDIs or FGDs. We used the convenient sampling method based on the different categories of actors we aimed to represent at each site and decision-making level, but not according to their numbers. Theoretically, the sample size is guided when the data saturation is reached among our studied population. See highlighted in yellow lines 200 to 203 in the track changes version and lines 170 to 173 unmarked version.

10. Material and methods : It would be good to provide, possibly in supplementary materials what are the functions of the different groups interviewed (what does their functions actually involve on a daily basis).

Authors’ response: We thank reviewer 3 for this comment. A supporting information file has been added to describe stakeholders, organizations and roles in detail (S4).

11. Material and methods : On the description of the IDI and FDG :

• Design of table 1 would benefit from redesign to make it easier to read. At least repeat heard row on each page of the table.

Authors’ response: We have created a new table (lines 237 to 245 track changes version and lines 206 to 2013 unmarked version).

• Why are there IDIs with several interviewees cf. IDI 2, 3,4, 5 and 6? Was the same guide used for IDI/FDG or was it different thematic. It is not really clear when looking at the table.

Authors’ response: There were two guides: one for IDIs and another for FGDs. Both guides are presented in the supporting information (S1 and S2).

• As it reads in table 1, 93 participants were interviewed in Phase 1, 3 in Phase 2 and 64 in phase 3. You may want to double check the difference between number participants for phase 1 mentioned on line 183 (91) to see if there was not a typo in the table.

Authors’ response: Thank you for your vigilance. We have double checked and updated the table.

• In Table 1, data are presented for 11 IDI and FDG. It is explained in lines 199-201 that 2 IDI and 2 FDG were used to validate the tool and that it included xx participants. Table 1 reaches a total of 160 participants when “only” 158 are mentioned, and if we removed the 12 participants from the 2 IDI and FDG, then it should only be 146 participants in table 1. Please review the discrepancy.

Authors’ response: Thank you for your vigilance. We have double checked and corrected the discrepancies.

• Please explain what is the difference between phase 1 and 2? Was this because interviewees interviewed in phase 2 were not available in phase 2?

Authors’ response: The phase 1 has been implemented at the local level in Forest Guinea whereas the phase 2 has been implemented at the national level in Conakry several months after the first phase (partly due to the availability and the need to schedule interviews with national stakeholders in advance).

• One line 207 it is mentioned that FDG had 6 to 12 participants, please explain why several FDG had a total of more participants cf. FDG 1, 3, 8, 9 etc. – or was it several FDG on the same topic and/or with same target group?

Authors’ response: Indeed, the way the data was presented in the old version of the table led to inconsistencies. These inconsistencies are no longer present in the new version of the table (lines 237 to 245 track changes version and lines 206 to 2013 unmarked version).

• Would the authors consider providing the interview guides (IDI and FDG) in supplementary material?

Authors’ response: We added the interview guides to supplementary materials S1 and S2.

• Lines 210-212 : please consider rephrasing sentence as it is not clear in current format and that later in the limitation there is some discussions as of why less women were included and how it could impact perception. It is not clear in these lines whether or not authors consider that gender in-balance may be a limitation or not.

Author’s response: We understand the confusion you raised. In the Guinean context among Muslims, women typically do not feel at ease expressing their own opinions when men are also present in discussions. As a result such, we should have organized separate discussion groups: groups with only men and groups with only women. However, when these discussions happen in the socioprofessional sphere, this cultural sensitivity does not prevent both genders from freely discussing and expressing their points of view. Mixed-gender groups were not perceived as a limitation in that regards. To clarify this point, the sentences now read as: “The groups were homogeneous in terms of stakeholder categories (i.e., community members, community workers, local staff, prefectural staff, and national service staff), but not necessarily in terms of gender. This was because the presence of men in the same group did not prevent women from expressing their opinions. Based on knowledge of the local context, gender-related cultural sensitivity does not prevent individuals in mixed-gender group discussions from freely expressing their points of view when in a socioprofessional sphere ..” (highlighted in yellow lines 270 to 276 track changes version and lines 227 to 230 unmarked version).

Because so few women are officially involved in that field, they were de facto underrepresented, which could possibly have impacted and limited the diversity of viewpoints, hence our results. We added information to explain this limitation: “The low participation rate of women in the study can be explained by the fewer number of women working in technical services, such as veterinary services, at national level. Thus, our results need to be taken with caution as the management of alert and response mechanisms might be different if more women were included in our study as informal and nontechnical actors involved in alert and response management. (highlighted in yellow lines 557 to 561 track changes version and lines 474 to 478 unmarked version).

• You may want to consider providing a definition to “central”, “deconcentrated” and “local”. Also in the document, “deconcentrated” and “decentralized” seem to be used interchangeably cf. lines 253, 263, 334-335– these terms are not synonyms or if used in a similar way this should be explained and defined.

Authors’ response: We thank reviewer 3 for this comment. A supporting information file has been added to more clearly describe stakeholders, organizations and roles (S4). We also harmonized the terminology use throughout the document.

12. Results : Lines 261-268 : it is mentioned that results will be presented through 5 themes. Last theme “perception of control measures” should include a headline similarly to the other themes and be developed on the result section – or be removed from text. It currently does not appear.

Authors’ response: We thank reviewer 3 for this comment. We have removed “perception of control measures” from the included themes.

13. Results : It would be good to describe the weighting of the themes that came out from the IDI and FDG. How were they classified and weighted to come out with the one selected. It’d be good as well to quantify (as possible) how they were perceived by category of interviewees (for ex. 80% of x group reported issues with access to resources for instance).

Authors’ response: There are indeed several schools of thought when it comes to qualita

---

## [Decision Letter · Decision Letter 1]

6 Aug 2025

Rapid response to hemorrhagic fever emergence in Guinea: community-based systems can enhance engagement and sustainability

PONE-D-25-09977R1

Dear Dr. Olive,

We’re pleased to inform you that your manuscript has been judged scientifically suitable for publication and will be formally accepted for publication once it meets all outstanding technical requirements. Please consider addressing the last comment made by referee #3, it should not take you long to do this.

Kind regards,

Ted Loch-Temzelides

Academic Editor

PLOS ONE

Reviewers' comments:

Reviewer's Responses to Questions

**Comments to the Author**

Reviewer #1: All comments have been addressed

Reviewer #3: All comments have been addressed

2. Is the manuscript technically sound, and do the data support the conclusions?

Reviewer #1: Yes

Reviewer #3: Yes

3. Has the statistical analysis been performed appropriately and rigorously?

Reviewer #1: N/A

Reviewer #3: Yes

4. Have the authors made all data underlying the findings in their manuscript fully available?

Reviewer #1: Yes

Reviewer #3: Yes

5. Is the manuscript presented in an intelligible fashion and written in standard English?

Reviewer #1: Yes

Reviewer #3: Yes

Reviewer #1: This is important work and highlights how important community engagement is when addressing disease outbreaks. The article reads more clearly now.

Reviewer #3: The reviewer thanks the authors for the work done in revising the manuscript. It reads really well and provides great insights to this important topic. All comments raised were adequately answered to, this is very much appreciated. English language has been reviewed and improved. The authors also provided supplementary materials that complement nicely the manuscript. Really well done.

Please see below a minor comment you might want to add in the final version.

Lines 102-104 “affected communities often did not abide by measures such as (…) vaccination of contact cases and at-risk individuals”. You may want to recall that vaccination was, at that time, conducted through a clinical trial and was not widely available throughout the outbreak (trial started on March 2025) – it seems a bit unfair, to state that affected communities did not abide by this specific measure. A for all trial, informed consent is a must. I would suggest, either to remove this “measure” or to clearly state that this was a clinical trial that started a year after the outbreak was declared.

---

## [Editor Report · Acceptance letter]

PONE-D-25-09977R1

PLOS ONE

Dear Dr. Olive,

I'm pleased to inform you that your manuscript has been deemed suitable for publication in PLOS ONE. Congratulations! Your manuscript is now being handed over to our production team.

Kind regards,

on behalf of

Dr. Ted Loch-Temzelides

Academic Editor

PLOS ONE